# Antiviral Activity of a Turbot (*Scophthalmus maximus*) NK-Lysin Peptide by Inhibition of Low-pH Virus-Induced Membrane Fusion

**DOI:** 10.3390/md17020087

**Published:** 2019-02-01

**Authors:** Alberto Falco, Regla María Medina-Gali, José Antonio Poveda, Melissa Bello-Perez, Beatriz Novoa, José Antonio Encinar

**Affiliations:** 1Instituto de Investigación, Desarrollo e Innovación en Biotecnología Sanitaria de Elche (IDiBE) and Instituto de Biología Molecular y Celular (IBMC), Miguel Hernández University (UMH); 03202 Elche Alicante, Spain; rmedina@umh.es (R.M.M.-G.); ja.poveda@umh.es (J.A.P.); mbello@umh.es (M.B.-P.); 2Instituto de Investigaciones Marinas (IIM), Consejo Superior de Investigaciones Científicas (CSIC), 36208 Vigo, Spain; beatriznovoa@iim.csic.es

**Keywords:** NK-lysin, Nkl_71–100_, phospholipid vesicles, aggregation, leakage, phosphatidylserine, antiviral, viral fusion, SVCV

## Abstract

Global health is under attack by increasingly-frequent pandemics of viral origin. Antimicrobial peptides are a valuable tool to combat pathogenic microorganisms. Previous studies from our group have shown that the membrane-lytic region of turbot (*Scophthalmus maximus*) NK-lysine short peptide (Nkl_71–100_) exerts an anti-protozoal activity, probably due to membrane rupture. In addition, NK-lysine protein is highly expressed in zebrafish in response to viral infections. In this work several biophysical methods, such as vesicle aggregation, leakage and fluorescence anisotropy, are employed to investigate the interaction of Nkl_71–100_ with different glycerophospholipid vesicles. At acidic pH, Nkl_71–100_ preferably interacts with phosphatidylserine (PS), disrupts PS membranes, and allows the content leakage from vesicles. Furthermore, Nkl_71–100_ exerts strong antiviral activity against spring viremia of carp virus (SVCV) by inhibiting not only the binding of viral particles to host cells, but also the fusion of virus and cell membranes, which requires a low pH context. Such antiviral activity seems to be related to the important role that PS plays in these steps of the replication cycle of SVCV, a feature that is shared by other families of virus-comprising members with health and veterinary relevance. Consequently, Nkl_71–100_ is shown as a promising broad-spectrum antiviral candidate.

## 1. Introduction

Nowadays, community health is under attack by the continuous emergence of deadly viral outbreaks that result in serious pandemics such as the quite recent seasonal flu virus A (H1N1), avian influenza virus A (H5N1) and Ebola virus (EV) [1]. Despite recent technological and biomedical advances, the global burden of morbidity and mortality arising from viral infections is disproportionately high, especially in children [2]. The reasons that contribute to this scenario are diverse: (i) inefficient vaccines [3,4], (ii) stricter safety requirements for vaccines than for other drugs [5], (iii) focused efforts against the acquired immunodeficiency syndrome (AIDS) to the detriment of other viral diseases [6,7], (iv) the appearance of new viral pathogenic strains [1], (v) and resistance phenomena, among others. Therefore, renewed efforts to discover new broad-spectrum antiviral drugs are underway [1,7].

Although the described situation might appear disappointing, there is confidence that new antiviral drugs will be available to add to the limited arsenal of antivirals and to support medicine in keeping/improving current health standards. Significant advances, by means of increasingly-potent bioinformatic tools and associated databases, are being achieved through the screening of large compound libraries for targeting critical specific sites within functionally-relevant proteins [8]. Other approaches to increase the discovery of new antivirals focus on the study of nature, particularly with respect to the innate immune system, since the ancestral battle for supremacy between virus, prokaryotes and eukaryotes have developed and cumulated a collection of optimized antimicrobial strategies and molecules for millions of years.

In this regard, antimicrobial peptides (AMPs) play the role of effectors of the innate immune system in all existing types of organisms, and hence, they have been shaped for this task by the selective pressure of host-pathogen interactions. As a result, AMPs can be considered as multiple impact weapons that comprise direct and diverse strong activity against a broad range of microbes, as well as potent immunomodulatory substances, which together contribute to making the development of resistant pathogens [9,10,11] difficult. In general, the cationic and amphipathic nature of AMPs allows them to interact with the commonly negatively-charged lipid cell membranes of pathogens. Thus, the proposed mechanisms include different pore-formation models that finally induce the destabilization/permeabilization of their membranes and, consequently, the killing of the microorganism [12,13].

AMPs show extraordinary diversity in sequence and structure, and it is believed that fish may possess a broader range of AMPs and other antimicrobial molecules in their mucosal surfaces than “higher” vertebrates, since they rely heavily on the innate immune responses to combat infections [14,15]. Indeed, lots of different AMPs, some with reported antimicrobial (including antiviral) activity, have been identified within a broad spectrum of different fish species, belonging not only to families of AMPs exclusive to fish, for instance, piscidins, pleurocidins and chrysophsins, but also present in different groups of vertebrates and invertebrates, such as histone H2A-derived peptides, defensins, hepcidins, cathelicidins and NK-lysins [16,17].

Among them, research on NK-lysins is recently attracting special attention, as they have been found to exert significant broad-spectrum activity against not only all kind of microorganisms (i.e., Gram-negative and -positive bacteria [18,19,20,21,22,23,24], virus [25], fungi [22] and protozoa [22,26,27,28]), but also tumor cells [29]. Since first reported in porcine cytotoxic lymphocytes in 1995 [30], NK-lysin orthologs have been identified in all vertebrate groups [31,32,33] with structural homology with saposin-like proteins (SAPLIPs), surfactant protein B (SP-B), amoeba pore-forming polypeptides (amoebapores), the cyclic peptide bacteriocin AS-48 and some regions of prophytepsin, acid sphingomyelinase and acyloxyacyl hydrolase. All of them are members of the “saposin fold” family, whose functionalities are associated to lipid interactions [34,35,36,37,38,39,40]. Thus, the commonly over 100 aa residues of NK-lysin mature peptides are structured in a five-helix bundle which is stabilized by three intrachain disulphide bonds between their six half-cystine residues. In such conformation, there is one face of the molecule with net positive charge whose arginine residues may mediate the membrane binding and lysis [41]. Actually, this region, which comprises both contiguous and disulphide-bonded helices 2 and 3, has been identified as the most membrane-lytic portion of the sequence as reported for several peptides derived from this region [42,43,44,45,46,47].

Homologues of NK-lysin have been discovered in several fish species [19,20,21,27,31,48,49,50], but functional assays have mainly focused on their antibacterial activity [18,19,20,21,50]. A recent study on turbot (*Scophthalmus maximus*) NK-lysin suggests a relevant role in antiviral immunity as their transcript levels significantly increased in fish main immune organs, i.e., head kidney and spleen, in response to the experimental in vivo infection with viral hemorrhagic septicemia virus (VHSV) [51]. Additionally, constitutive levels of NK-lysin transcripts in head kidney showed to be significantly higher in VHSV-resistant strains of turbot than in VHSV-susceptible ones [51].

In a previous report of our group, we showed that the peptide Nkl_71–100_ presents antiparasitic activity inhibiting the proliferation of *Philasterides dicentrarchi*, which is probably dependent on the rupture of its membrane [27]. The present study aims to demonstrate the specific interaction of the Nkl_71–100_ peptide with glycerophospholipids using biophysical techniques. The biophysical data presented here show its preferential interaction with PS at acidic pH. This initial observation allowed us to suppose that the peptide Nkl_71–100_ could interfere in the replicative cycle of viruses whose entry into the host cell implies a process of receptor-mediated endocytosis, followed by another fusion step to the lysosome membrane at acidic pH. Therefore, our second main objective has been to demonstrate the antiviral activity of the peptide Nkl_71–100_ by using spring viremia of carp virus (SVCV) as model rhabdovirus.

## 2. Results

### 2.1. Interaction of Nkl_71–100_ with Membranes

Changes in the absorbance at 360 nm after mixing phospholipid vesicles with Nkl_71–100_ at different phospholipid/peptide ratios are shown in Figure 1. Since there are no absorbing species at 360 nm in either the phospholipids used or Nkl_71–100_, the apparent changes in absorbance are probably due to the scattered light produced by the size increase or aggregation of the phospholipid vesicles because of the interaction with Nkl_71–100_. At pH 7, such interaction is almost negligible, independent of the phospholipid used; however, at acidic pH, the absorbance increases in all cases, but it is especially remarkably for PS vesicles, indicating specificity for this anionic phospholipid. Note that the scale of the axis-ordinates is different between Figure 1A (PC and PG vesicles) and 1B (PS vesicles).

The destabilization of PS bilayers by Nkl_71–100_ was further assessed by determining its ability to induce aqueous content leakage from PS vesicles as previously described by Ellens et al. (1985) [52]. Figure 2 shows the aqueous content leakage induced by different concentrations of Nkl_71–100_ (0–24 µM) in a PS vesicle suspension at pH 3 and 7. As expected, Nkl_71–100_ was able to induce the release of the vesicle content, which was significantly higher at pH 3, in a concentration-dependent manner. Such release plateaued at both neutral (≈20%) and acidic (≈50%) pH at about 8 and 12 µM of peptide, respectively. Irrelevant changes in the leakage of aqueous content were detected using PC or PG lipid vesicles, both at pH 3 and pH 7 (data not shown), in agreement with the low levels of aggregation shown in Figure 1.

### 2.2. Studies of Peptide Insertion into Phospholipid Bilayers

Previous results reveal a specific interaction between the Nkl_71–100_ peptide and PS vesicles at pH 3, leading to a large vesicle aggregation (Figure 1) and content leakage (Figure 2). To further characterize this interaction, fluorescence anisotropy experiments of DPH-labelled DMPS vesicles were performed. DPH gets into the lipid bilayer in between the acyl chains, thus being sensitive to the fluidity of the inner membrane [53]. For these experiments, DMPS has been used instead of PS to measure the phase transition of this lipid, a parameter that helps to evaluate the peptide-lipid interaction. When DPH anisotropy was measured at different temperatures in control vesicles with no peptide added, an expected sigmoidal decay curve is obtained, reflecting the cooperative lipid phase transition from an ordered gel to a more fluid liquid-crystalline phase (Figure 3). The temperature of this transition (T_m_) is higher at pH 3 compared to pH 7, indicative of a reduction of the electrostatic repulsion between the DMPS polar headgroups, and thus, a stronger packaging of lipid molecules. This is in agreement with the expected protonation of this group, since the reported pK_a_ for this lipid is around 3.6 [54], then becoming a zwitterionic lipid at pH 3 instead of an anionic one when at pH 7. Regarding the effect of Nkl_71–100_, experiments were performed at 18.7 µM peptide concentration, a condition showing a maximum effect on vesicle aggregation and leakage. Although at both pHs the effect of the peptide on the DPH anisotropy curve is limited, there are some interesting differences. At pH 7, there is a slight increase in the T_m_ and a decrease of the anisotropy of DPH only in the gel phase. However, at pH 3, T_m_ slightly decreases and the drop in the anisotropy occurs at any temperature, that is, in the gel and the fluid phase. These small T_m_ variations suggest that although electrostatics could have a certain influence in the lipid-peptide interaction, explaining why the zwitterionic PC shows a lesser effect relative to vesicle aggregation, it should not be the main driving force for this interaction, in agreement with the little vesicle aggregation observed for the anionic lipid PG or PS (when at pH 7) compared to the zwitterionic PS (when at pH 3). Relative to the effect on the DPH anisotropy, this parameter is sensitive to the membrane fluidity mainly at the level of the acyl chains. The obtained results point to a difference in the interaction of the peptide and DMPS depending on the pH. Although still speculative, since a deeper biophysical study would be needed to draw more solid conclusions, it seems that at pH 3, the peptide could insert into the membrane interior in a stable way. In this position, it would not strongly perturb the headgroup packing (and thus T_m_), but would increase the acyl chains fluidity, then decreasing the DPH anisotropy. However, at pH 7, this effect seems to be limited to the gel phase, so at the fluid phase the peptide could be partly excluded or displaced from the membrane interior. This would explain the lower effect of the peptide at neutral pH on vesicle aggregation and leakage of PS, since this lipid stays at a fluid phase in those experiments.

### 2.3. Assessment of the Nkl_71–100_ Antiviral Activity

First, cellular cytotoxicity induced by Nkl_71–100_ was determined by means of an MTT cell-viability assay (Figure 4). For this purpose, EPC cells were treated with the peptide at different concentrations ranging from 0 to 64 µM for 24 h. No significant toxic effect was observed at concentrations lower than 32 μM (84.5 ± 1.9%), but at higher concentrations, viability dropped severely. Thus, the following antiviral assays were performed by using a maximum Nkl_71–100_ concentration of 32 µM.

The first approach to determine the antiviral activity of Nkl_71–100_ was by co-incubating the peptide (0–32 µM) with the SVCV infectious particles (MOI 10^−3^) prior to cell inoculation. The enveloped nature of such a virus makes it suitable to be targeted by Nkl_71–100_, as deduced from the membrane-interaction biophysical results. However, despite the high inhibition of the viral replication by the presence of the peptide in a dose-dependent manner that was observed, such an effect was independent of the preincubation period (Figure 5) (no statistically significant differences were found by multiple t test analysis among both groups). Fitting the parameters of a Hill equation for each data series, the estimated IC_50_ value for a preincubation period of 24 h (7.2 µM) was very similar to that obtained when treatments and virus were inoculated to cells at the same time, and just only concurring during the adsorption stage (t = 0, 8.1 µM), which suggests that the observed inhibition does not occur as a consequence of a direct effect of Nkl_71–100_ on the viral particles but during the adsorption period and/or later.

Thus, the subsequent efforts were directed towards ascertaining the stage of the viral cycle in which the peptide was exerting the inhibitory activity on SVCV replication. For such a task, treatments, consisting in the same range of concentrations of Nkl_71–100_ as in Figure 5, were added to cells at different critical points within viral infection, i.e., 2 h before the inoculation of the virus (pre-adsorption), and just after the adsorption period for either 2 or 20 h (post-adsorption) (Figure 6). The inhibition of the infectivity of SVCV occurred in a dose-dependent manner in all cases, but differed in the potency of the activity. At the maximum concentration tested (32 μM), the SVCV infection percentages were 47.1 ± 3.7, 20.1 ± 4.3 and 0% for the pre-adsorption, 0–2 h post-adsorption and 0–20 h adsorption treatments, respectively, indicating, together with previous results, that Nkl_71–100_ is at least (since potential inhibitory effects at late steps of infection cannot be discarded yet) affecting viral replication at an early stage in which virus-cell interactions are implicated, most probably at membrane levels.

### 2.4. Determination of the Antiviral Effect of Nkl_71–100_ on the Viral Entry Stage

In general terms, initial stages of the viral replication cycle comprise several steps, such as the attachment of the virions to the cell surface and the binding of viral surface proteins to the host’s specific receptors and the entry of the virions into the host cell, which in the case of an enveloped virus (as for instance SVCV) includes the fusion of viral and cellular membranes [55]. Therefore, in order to determine whether Nkl_71–100_ might be disrupting any of these mechanisms, following assays focused on the study of such events, both mediated in SVCV by the unique protein at its surface, the glycoprotein G (gpG). To study the influence of Nkl_71–100_ on the binding between the viral particles to host cell surfaces, EPC cells were inoculated with SVCV at MOI 1 together with 8 and 24 μM of peptide and incubated for 2 h at 4 °C to allow just the adsorption step. Afterwards, the inoculum mixtures were removed, and the cell monolayers washed to remove non-attached viral particles, while adsorbed ones were subsequently quantified by analyzing the abundance of both viral *n* and *g* gene copies (Figure 7). In this way, it was observed that Nkl_71–100_ alters the binding ability of SVCV to cell membranes since virus-cell binding levels significantly decreased in the presence of peptide in comparison to what it is observed in its absence. In addition, although only two different concentrations of Nkl_71–100_ were tested, such inhibition appears to occur in a concentration-dependent manner (~50% for 8 μM and ~61% for 24 μM).

However, even being significant, the level of binding inhibition exerted by Nkl_71–100_ might not be enough to explain previously-reported high antiviral activity. Moreover, the interference of the peptide the viral attachment ability does not discard its implication with other viral mechanisms, since such activity is very likely associated to the interaction of viral gpG with host cell components. Therefore, Nkl_71–100_ might still be playing an inhibitory role in other subsequent related mechanisms involving the same actors, such as the virus-cell membrane fusion.

This alternative was further studied by performing an SVCV-gpG-dependent fusion assay, which consists of lowering to pH 5–6 the cell culture media of SVCV-infected cells, and thus inducing the expression of the SVCV gpG at their membranes, in order to generate cell-to-cell fusion and the formation of quantifiable syncytia (multinucleated cells). By following this procedure, the influence of Nkl_71–100_ at 8 and 24 μM was assessed when either incubated with cells for 2 h and then removed prior to the triggering of the fusion mechanism, or just added at that time (Figure 8). Results indicate that Nkl_71–100_ is able to inhibit gpG-mediated fusion formation in both cases (~33% when preincubated 2 h and ~67% when added at the time of triggering the fusion, both at 24 μM), which not only suggests again that Nkl_71–100_ interacts with either virus or cell components since its activity remains even if removed from the system, but also that such interaction disturbs the fusion activity of SVCV gpG.

## 3. Discussion

Since their discovery [30], all studies aimed at describing the activity of NK-lysins point at their interaction with lipid membranes as a key factor [35,41,57,58]. In fact, these peptides share the “saposin fold” structural pattern with other molecular families whose activity is always associated with lipid interactions [34,35,36,37,38,39,40]. In the present study, we explore the affinity and interaction capacity with lipid membranes of a fragment, described in other orthologs such as the membrane-lytic region, of turbot NK-lysine (Nkl_71–100_). In fact, a previous study had already given experimental indications of such membrane-lytic activity by inducing the destructuring of the protozoan membrane of *Philasterides dicentrarchi*, a fish pathogen [27]. The biophysical tests (phospholipid vesicles aggregation, leakage and fluorescence anisotropy) carried out to characterize the interaction of Nkl_71–100_ with phospholipid membranes revealed that the peptide has a specific interaction with PS relative to PC or PG. The electrostatic interactions seem not to be the main factor, since the effect on the anionic lipid PG is scarce. Moreover, relative to PS, the effect is clearly conditioned by the pH, being remarkably greater at acidic pH when PS losses a negative charge becoming a zwitterionic lipid. This biophysical characterization also suggests that the peptide can get into the PS membrane bilayer at acidic pH, thus inducing vesicle aggregation and content leakage. Similar results have been reported previously with other NK-lysins and derived peptides [25].

The fact that NK-lysins exert their antimicrobial activity by at least disturbing the membrane of microorganisms becomes evident from the observed morphological changes in them [24,27,45]. However, to better infer potential pathogen’s targetability by those antimicrobials aiming at lipids, further characterization of microbial lipidomes is required [7,59]. In contrast, further advances have been made in the study of cancer in this field, revealing that one of the most important molecular hallmarks in apoptosis is the externalization of PS molecules, which are usually located on the inner side in healthy cells, thus becoming a wide-spread cancer cell biomarker, as well as a promising therapeutic target [60,61,62,63]. In this sense, both NK-lysin [64], as well as its membranolytic derived peptides [65,66,67,68], have shown remarkable cytotoxic activity against cancer cells with solid correlations with the presence of PS at their surfaces [65,66,68]. Interestingly, many pathogens including protozoan, bacteria and viruses use PS to camouflage themselves as apoptosis debris-like particles in order to evade the immune system and facilitate their infection [60]. In the specific case of viruses, this strategy termed as “apoptotic mimicry” is used by many virus families (including enveloped and non-enveloped viruses) to also facilitate their binding and entry [60,69]. Therefore, it is not surprising that some PS-targeting compounds, such as bavituximab, are being assessed clinically as therapeutic candidates for the treatment of both cancer and viral diseases [7,60].

In NK-lysins, the antiviral activity is much less characterized than the anticancer one. To our knowledge, there is only one study in which “direct” in vitro antiviral activity of NK-lysins is demonstrated [25]. In this study, it is shown that granulysin inhibits the infectivity of varicella-zoster virus (VZV) in a time and concentration-dependent manner in monolayers of infected MRC-5 cells (similarly as shown in Figure 6 of the present work); however, this study does not demonstrate activity on the viral particle or a specific process of its replication, but on the infected cells, in which infection induces apoptosis and the consequent exposure of PS molecules, which is also accelerated by the treatment with the peptide [25]. In any case, their activity on viral particles or some of the processes involved in their replication is potentially possible due to the presence of PS in the membranes involved, as shown in this work [7,59]. In addition, as it has been shown in fish, the fact that the expression of NK-lysins is stimulated against viral infections [48,49,51], as well as that their administration increases the levels of protection against viral infections in vivo [42,48], suggests a more important antiviral role of these peptides.

SVCV, like the rest of rhabdoviruses, is an enveloped virus, whose unique membrane protein (gpG) acts by binding to the specific cell receptor of the target cell, which induces an endocytic process that internalizes the viral particle within a lysosome from which it will later be released into the cytosol by the gpG-mediated fusion of the viral and endosome membranes. Therefore, within their life cycle, apart from the exit stage, these are the main processes (included in the viral entry phase) in which a compound that attacks membranes during virus-cell interactions could act [7,55]. Then, in this work, after verifying that Nkl_71–100_ possesses antiviral activity, and having discarded the direct effect on the viral particle, deduced from the almost identical inhibition profiles observed when virus is pretreated with the peptide or it is not (Figure 5), the potential effects of Nkl_71–100_ on the entry stage processes were studied. In addition, given that in this assay Nkl_71–100_ is only present in the early stages of the infection process (until the end of the virus adsorption period specifically), such inhibition must be due to the interference with entry processes mainly. The results shown in Figure 6 not only support this hypothesis, but also suggest that (i) the peptide interacts with cellular elements since certain antiviral activity is detected when just cells are pretreated for only 2 h prior to peptide removal and virus inoculation, and that (ii) the entry phase is affected since higher antiviral activity is observed when the peptide is only present during the adsorption period.

For these reasons, our next efforts were focused on determining if Nkl_71–100_ was affecting the adsorption/binding and/or fusion processes. In this sense, the high and pH-dependent affinity of Nkl_71–100_ for PS revealed in previous biophysical studies may shed some light. As already mentioned, the only protein of the rhabdoviruses involved in these two processes is gpG, which is widely described as interacting with cell membrane phospholipids [55,70] and, specifically, with PS [71,72,73]. In fact, other studies have shown that treatment with PS inhibits the adsorption of vesicular stomatitis virus (VSV) [74] and the binding of viral hemorrhagic septicemia virus (VHSV) [73], and consequently, their infection. This is due to the existence of a PS binding region in the gpG, conserved in all rhabdoviruses [75], and whose role in the infectivity is critical, as evidenced not only by the fact that specific antibodies against this region inhibits the fusogenic capacity of the gpG [76,77], but also since the peptides derived from this region (called PS-VHSVG-peptides from now on) preserve the affinity for PS and manage to inhibit the infectivity of the virus [78]. In this sense, biophysical studies with one of these peptides exhibit a very similar behavior to the one observed for Nkl_71–100_ in this work [79], suggesting, when taken all together, the presence of similar inhibitory mechanisms to induce gpG dysfunction when intervening in the interaction with PS. In previous work reporting the antiviral activity of the PS-VHSVG-peptides [78], these are only tested by removing them at the end of the adsorption, having been pre-incubated with the virus prior to its inoculation, as with the results shown Figure 5. In this work, peptides are tested at both acidic and physiological pH, observing that, unlike Nkl_71–100_, the inhibition of infectivity mediated by PS-VHSVG-peptides necessarily requires an acidic pH at the moment of its addition [80]. This fact might be explained because, although both peptides require an acidic pH to exert their lytic activity in membranes with PS, only Nkl_71–100_ is able to bind at neutral pH to another target related to virus-cell interactions, which is in symphony with the ability of Nkl_71–100_ to inhibit viral absorption and/or binding. In turn, once integrated into the system, the acidic pH necessary to trigger its binding to PS and membranolytic activity would occur naturally in the endocytic compartment; an interesting fact for clinical application assessments, since such selective and local activation may presumably contribute to the reduction of potential toxicity issues.

Altogether, the importance of this discovery lies in the fact that Nkl_71–100_ targets common but essential viral functions (i.e., attachment/binding and fusion) that are closely related to a structural component, i.e., the membrane, making it a promising broad-spectrum antiviral for which it is also difficult to develop resistance. Since most relevant viral pathogens are membrane-enveloped viruses that require membrane fusogenic proteins to access the host’s cytosol, the quantity of potential important targets becomes substantial. Besides, among the three most-common structurally-distinct classes of viral fusion proteins, only some class I fusion protein members, such as the human immunodeficiency virus (HIV), are functionally pH-independent, and thus, its replication could not be blocked by this action mode, but maybe by alternative ones. In any case, by specifically aiming at low-pH induced fusion, the Nkl_71–100_ spectrum of potential viral targets comprises, for instance, influenza (Class I), flaviviruses such as Hepatitis C (HCV), Dengue (DENV), Zika (ZV) and West Nile (WNV) viruses (Class II) and rhabdoviruses and herpesviruses (Class III) [7]. All this invites us to guide our future work around determining the spectral range of the antiviral activity of this peptide and its derivatives, which could break the old paradigm “one bug, one drug”.

## 4. Materials and Methods

### 4.1. Reagents

In this work, the peptide from aa residue 71 to 100 that comprises the membrane-lytic region of the turbot NK-lysin (Nkl_71–100_) reported in Lama et al. (2018) [27] was used (^71^EGVKSKLNIVCNEIGLLKSLCRKFVNSHIW^100^). The synthetic peptide was purchased from ShineGene (Shanghai, China) with a purity grade >95%.

Egg phosphatidylcholine (PC), egg L-α-phosphatidylglycerol (PG) and bovine brain phosphatidylserine (PS) and 1,2-dimyristoyl-sn-glycero-3-phospho-L-serine (DMPS) were provided by Avanti Polar Lipids (Alabaster, AL, USA). Carboxyfluorescein, 1,6-Diphenyl-1,3,5-hexatriene (DPH), thiazolyl blue tetrazolium bromide (MTT), formalin, triton X-100 and chloroform were obtained from Sigma-Aldrich (Sant Louise, MO, USA). Anhydrous dimethyl sulfoxide (DMSO) was purchased from Merck (Kenilworth, NJ, USA).

### 4.2. Large Unilamellar Vesicles (LUVs)

For the preparation of LUVs vesicles, a phospholipid film was obtained upon overnight drying of a chloroform solution under vacuum. The phospholipids were resuspended at a concentration of 1 mg/ml in medium buffer (100 mM NaCl, 5 mM MES, 5 mM sodium citrate, 5 mM Tris, 1 mM EDTA) at the appropriate pH value (either 3 or 7) for 1 h at 37 °C (or above the lipid phase transition temperature (T_m_) for DMPS) and vortexed vigorously. This suspension was subjected to 19 cycles of extrusion in a LiposoFast-Basic extrusion apparatus with polycarbonate filters of 100 nm (Avestin, Ottawa, Canada). For DMPS samples, a pressure extrusion system was used with the same filters, thermostatized above the lipid T_m_. A 0.14 mM phospholipid final concentration was used in all experiments.

### 4.3. Vesicle Aggregation Assay

The optical density (OD) variation at 360 nm produced by the addition of Nkl_71–100_ (final concentration range: 0–25 µM) to a LUV vesicle suspension in medium buffer at either pH 3 or 7 was measured on a Beckman DU-7 spectrophotometer (Beckman Instruments, Inc., Palo Alto, CA, USA) after 1 h incubation at 37 °C. To account for the absorbance of phospholipid vesicles and peptide alone, control samples containing equal amounts of DMSO, but in the absence of peptide or phospholipid, were measured.

### 4.4. Leakage Assay

The assay for leakage of liposomal contents was performed using the carboxyfluorescein assay [81]. Vesicles loaded with carboxyfluorescein were prepared by extrusion through a 100 nm filter (Avanti Mini-Extruder) of the PS lipid in a suspension containing 40 mM CF. Free CF was removed by passing the suspension through a Sephadex G-75 column (Pharmacia, Uppsala, Sweden) using 100 mM NaCl, 5 mM MES, 5 mM sodium citrate, 5 mM Tris, 1 mM EDTA at the desired pH value (either 3 or 7) as the elution buffer. Aliquots of eluted liposomes were used to determine the amount of phospholipid by the phosphorus assay. Changes in fluorescence intensity due to the presence of the peptide were recorded with excitation and emission wavelengths set at 492 nm and 520 nm, respectively. Leakage was initiated by the addition of the Nkl_71–100_ peptide (final concentration range: 0–24 µM) at either pH 3 or 7. 0.5% Triton X-100 was used as the total rupture control for the release of all trapped fluorescent dye. Leakage was quantified on a percentage basis according to the equation, % Release = 100 (F_f_ − F_0_)/(F_100_ − F_0_), F_f_ being the equilibrium value of fluorescence after peptide addition, F_0_ the initial fluorescence of the vesicle suspension and F_100_ the fluorescence value after the addition of Triton X-100 [81].

### 4.5. Fluorescence Anisotropy

Fluorescence anisotropy measurements of DPH were made on a Varian Cary Eclipse spectrofluorometer equipped with a Peltier thermostat system (Varian, Palo Alto, CA, USA). The DPH probe in *N*,*N*-Dimethylformamide was added to DMPS LUVs at a lipid/probe molar ratio of 500 at the appropriate pH and heated above the Tm of the lipid for 30 min to facilitate the probe incorporation into the LUVs. Peptide from a freshly-prepared stock solution in water was added to DMPS LUVs containing DPH, so that the final lipid and protein concentrations were 0.14 mM and 18.7 µM respectively. The excitation wavelength was set at 365 nm, while the emission was 425 nm. The temperature up-scan rate was set to 0.6 °C/min in all the experiments. The parameters of a sigmoid curve have been adjusted to the experimental data to calculate the transition temperature of the lipid phase using the GraphPad Prism 6 (GraphPad Software, Inc., La Jolla, CA, USA) software.

### 4.6. Cell lines and Virus

The *Epithelioma papulosum cyprinid* (EPC) cells from the fat-head minnow fish (*Pimephales promelas*) were obtained from the American Type Culture Collection (ATCC, Manassas, Vi, USA, code number CRL-2872). EPC cell monolayers were grown at 28 °C in a 5% CO_2_ atmosphere in Dutch modified Roswell Park Memorial Institute (RPMI) 1640 medium containing 10% FBS (Sigma-Aldrich), 1 mM sodium pyruvate, 2 mM glutamine and 50 μg/mL gentamicin and 2 μg/mL of fungizone (Gibco BRL-Invitrogen, Carlsbad, CA, USA).

For in vitro infections, SVCV isolate 56/70 from carp (*Cyprinus carpio*) [82] was replicated in EPC cell monolayers at 22 °C in an atmosphere without CO_2_ and the same culture media described above except for 2% FBS (infection media) [83]. SVCV-infected EPC supernatants were clarified by centrifugation at 4 °C and 4000 g for 30 min and stored at −80 °C until use.

### 4.7. Cytotoxicity Assays

The potential toxicity of Nkl_71–100_ on EPC cells was analyzed by measuring changes in cell viability by MTT assays as described previously [8]. Briefly, confluent cell monolayers in 96-well plates were treated with different concentrations (0–64 µM) of the peptide in infection media for 24 h (100 μL/well). Then, 0.5 mg/mL MTT from ten-fold concentrated stocks in PBS (stored at −20 °C) in fresh media (100 μL/well) was used to replace treatments. MTT solutions were incubated with cells under the same conditions for 4 h more. Finally, media were carefully removed, and the colored formazan product dissolved in 100 μL of DMSO. Then, absorbance at 570 nm and 620 nm as reference were measured by means of a SpectroStar Omega absorbance microplate reader (BMG LabTech, Ortenberg, Germany). Optical density is expressed in percentages relative to the control group consisting of untreated cells. Cell viability was calculated by the formula: 100× treated cell absorbance / untreated cell absorbance. All experiments were performed in tetraplicate, and results are shown as mean with standard deviation (s.d.) calculated from two different experiments.

### 4.8. Microscopy

The fluorescence and bright-field microscopy images were recorded by using a digital camera DFC3000G on a DMI 3000B inverted microscope with EL6000 compact light source, all of them from Leica (Bensheim, Germany). Images were acquired using either a 10× or 20× objective, and the fluorescein isotiocyanate (FITC) filter (Ex. BP 480/40, Em. BP 527/30) for the fluorescence ones. Leica Application Suite AF600 Module Systems was used for the manual formatting and processing of data acquisition.

### 4.9. In vitro Viral Infections

All in vitro viral infections included a 2 h adsorption step at 4 °C of the viral supernatants to the confluent EPC cell monolayers grown in 96-well plates, followed by 3 washes with infection media to eliminate unbound virus and further incubation at 22 °C for 20 h. Depending on the experiment, treatments were added at different stages of the viral infection process in order to determine when the antiviral activity of the peptide was significantly different. The multiplicity of infection (MOI) used for all experiments was 10^−3^. Once the EPC cell monolayers were infected, they were fixed with 4% formalin solution in PBS and infection levels quantified by immune-labeling of the infection foci as described elsewhere [84]. Briefly, fixed cells were incubated for 24 h at 4 °C with a polyclonal anti-SVCV (BioX Diagnostics SA, Jemelle, Belgium) diluted 1:300 in Ab-dilution buffer consisting in phosphate buffered saline (PBS) with 1% BSA, 1% goat serum and 0.5% Triton X-100. After 3 washes with PBS, cell monolayers were incubated for 45 min at room temperature with a FITC-labeled goat anti-mouse antibody (Sigma) diluted 1:300 in Ab-dilution buffer. After 3 more washes with PBS, stained SVCV-infection foci were then counted and photographed by means of the described-above microscope. SVCV infection was calculated in percentages by the formula: 100× number of foci obtained in treated infections/number of foci obtained in non-treated (control) infections. Three different assays, each in triplicate, were performed per experiment, and the results are shown as their mean ± s.d.

### 4.10. Viral Binding Assays

To determine whether Nkl_71–100_ inhibits the binding of SVCV particles to cells, EPC cell monolayers, grown in 96-well plates, were incubated with SVCV supernatants (MOI 1) in the absence or presence of the peptide at two concentrations (8 and 24 µM) for 2 h at 4 °C to allow virus binding/attachment but not its endocytosis. Cells were then washed 3 times with PBS to remove unbound virus, and cell-bound virus was then detected by reverse transcriptase quantitative polymerase chain reaction (RT-qPCR).

### 4.11. RNA Isolation, cDNA Synthesis and qPCR

For RNA extraction, the E.Z.N.A.^®^ Total RNA Kit I (Omega Bio-tek Inc., Doraville, GA, USA) was used following the manufacturer´s instructions. RNA samples were stored at −80 °C until use. For the synthesis of cDNA, one microgram of purified RNA from each sample was used, as estimated by a Nanodrop 1000 spectrophotometer (Thermo-fisher Scientific, Waltham, MA, USA), and the Moloney murine leukemia virus RT (Invitrogen), as previously described [85]. qPCR was then performed using an ABI PRISM 7300 thermocycler (Applied Biosystems, Foster City, CA, USA) with SYBR Green PCR master mix (Life Technologies, Paisley, United Kingdom). Reactions were prepared in a 20 μL total volume and included 2 μL of cDNA, 900 nM of each forward and reverse primers *(ef1a*, GeneBank Accession Number (GB acc. no.): AY643400.1, Fw: CTGGAGGCCAGCTCAAACAT, Rv: CATTTCCCTCCTTACGCTCAAC; *n_svcv_*, GB acc. no.: U18101, Fw: AGCTTGCATTTGAGATCGACATT, Rv: GCATTATGCCGCTCCAAGAG; *g*_svcv_, GB acc. no.: Z37505.1, Fw: GCTACATCGCATTCCTTTTGC, Rv: TCGACCAGATGGAACAAATATGG) and 10 μL of SYBR Green PCR master mix. Non-template controls were included for each gene analysis. Cycling conditions were 95 °C for 10 min, followed by 40 cycles of 1 min at 65 °C, 1 min at 95 °C, and finally, an extension of 10 min. Results were obtained by normalizing the number of each target gene (*g_SVCV_* and *n_SVCV_*) respective to that of the endogenous reference (transcripts of *ef1a*) in the same sample using a variation of Livak and Schmittgen’s method [80] by the formula 2^Ctref-Cttarget^. All reactions were performed in duplicate. Results are presented as percentages of normalized number of viral genome amplified fragments with corresponding specific primers in treated samples, relative to equivalent values in non-treated samples. Three different assays, each in duplicate, were performed per experiment, and the results are shown as their mean ± s.d.

### 4.12. Fusion Assays

Fusion assays were performed as previously described by Falco et al. [56]. Briefly, after the infection of EPC cells with SVCV as described above in section “In vitro viral infections”, the membrane fusion conformation of the G_svcv_ glycoprotein present at the membrane of infected cells was induced, which was waiting for the “budding” process of the viral replication cycle. In order to induce such conformation change and trigger the fusion of the membrane of infected cells with those of the surrounding cells to generate quantifiable syncytia, after the viral replication period of 20 h at 22 °C, cell media was removed, and cell monolayers washed thrice with infection media and then incubated with fusion media (i.e., infection media at pH 6) for 30 min at 22 °C. Afterwards, cell monolayers were washed and subsequently incubated with fusion media at pH 7.5 for 2 h at 22 °C. Within this experimental design, treatments of two concentrations of nkl_71–100_ (8 and 24 µM) were added either for 1 h after the replication period and prior to the incubation with pH 6 infection media or together with pH 6 infection media. Finally, cells were fixed with cold methanol (−20 °C) for 15 min, dried and stained with Giemsa (5 mg/mL in PBS). Syncytia were then counted and photographed with an inverted fluorescence microscope. G_svcv_-mediated syncytia production was calculated in percentages by the formula: 100× number of syncytia obtained in treated monolayers/number of syncytia obtained in corresponding non-treated (control) cell monolayers. Two different assays, each one in triplicate, were performed per experiment, and the results are shown as their mean ± s.d.

### 4.13. Statistical Analysis

Values are represented as the mean ± s.d. The data were subjected to statistical analysis: multiple t-test for the antiviral assays, one-way ANOVA followed by Dunnett′s multiple comparisons test for binding assays and two-way ANOVA followed by Dunnett’s multiple comparisons test for fusion assays. The differences were considered statistically significant at *p* < 0.05. All analyses were performed using the GraphPad Prism 6 (GraphPad Software, Inc., La Jolla, CA, USA). * *p* < 0.05, ** *p* < 0.01, and *** *p* < 0.001 on the bars, indicate statistically significant differences, compared to corresponding control groups.

## Figures and Tables

**Figure 1 marinedrugs-17-00087-f001:**
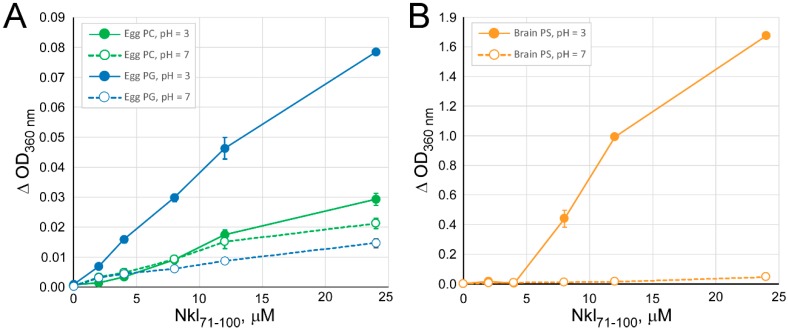
Vesicle aggregation induced by the interaction of the Nkl_71–100_ with phospholipid vesicles. To 1 ml of vesicles (0.14 mM) of PC (**A**), PG (**A**) or PS (**B**) in medium buffer at either pH 3 (full circles and continuous lines) or pH 7 (empty circles and dash lines), different amounts of Nkl_71–100_ were added to reach a final concentration of peptide ranging 0–25 µM. The absorbance was measured after incubating the sample for 1 h at 37 °C and presented as its variation respect to solvent control values. Results are shown as mean ± s. d. from three different experiments.

**Figure 2 marinedrugs-17-00087-f002:**
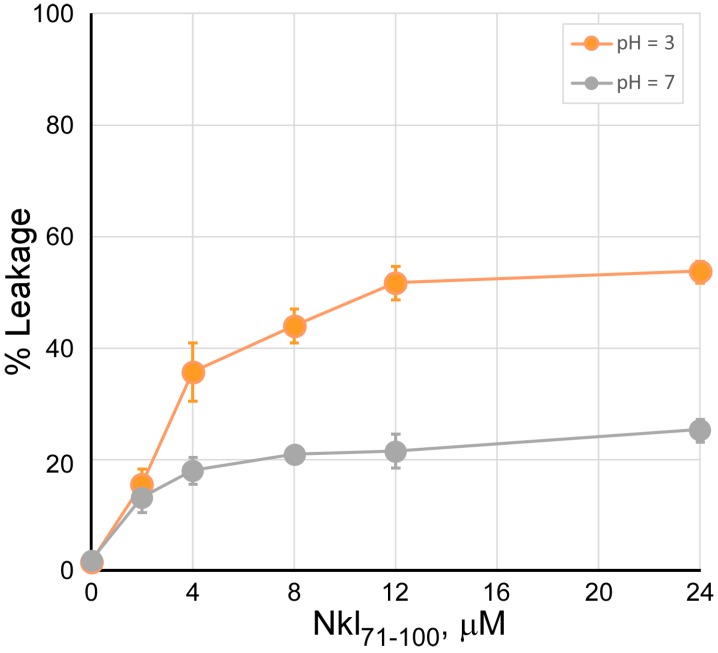
Carboxifluorescein leakage from phospholipid vesicles induced by Nkl_71–100_. To 100 µL of carboxifluorescein loaded PS vesicles (0.14 mM) in medium buffer at either pH 3 (orange) or pH 7 (gray), different amounts of Nkl_71–100_ were added to reach a final concentration of peptide ranging 0–24 µM. The mixtures were incubated at 37 °C for 30 min and the fluorescence intensity was recorded with excitation and emission wavelengths set at 492 nm and 520 nm, respectively. Results are shown in percentage (mean ± s. d., *n* = 3), considering maximal leakage that obtained upon addition of 0.5% Triton X-100.

**Figure 3 marinedrugs-17-00087-f003:**
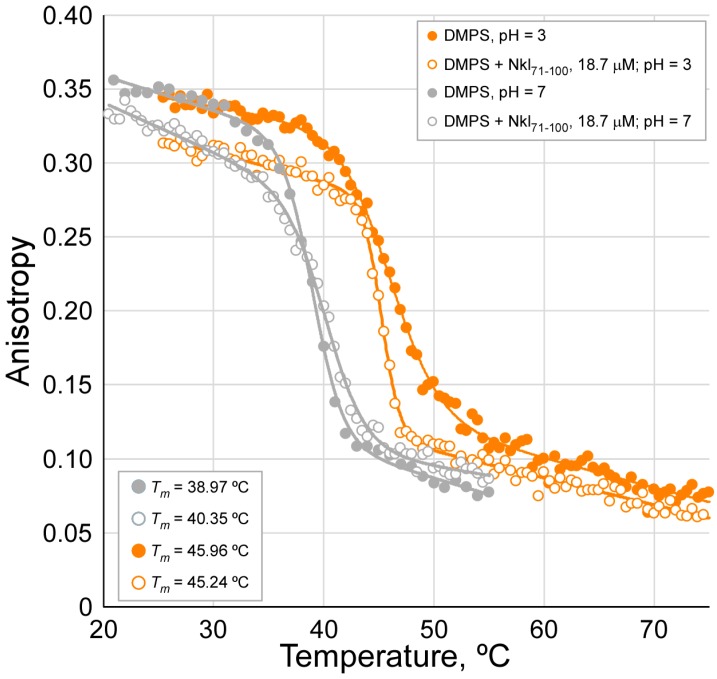
Effect of Nkl_71–100_ on the thermotropic behavior of DMPS vesicles. DPH-labeled DMPS LUVs vesicles, with or without added Nkl_71–100_ peptide, were submitted to a temperature ramp, and the anisotropy of the fluorescent probe was measured. The phospholipid concentration was 0.14 mM, the peptide 18.4 µM, and the DPH probe to lipid molar ratio 1 to 500. The lipid phase transition (T_m_) for each condition is indicated.

**Figure 4 marinedrugs-17-00087-f004:**
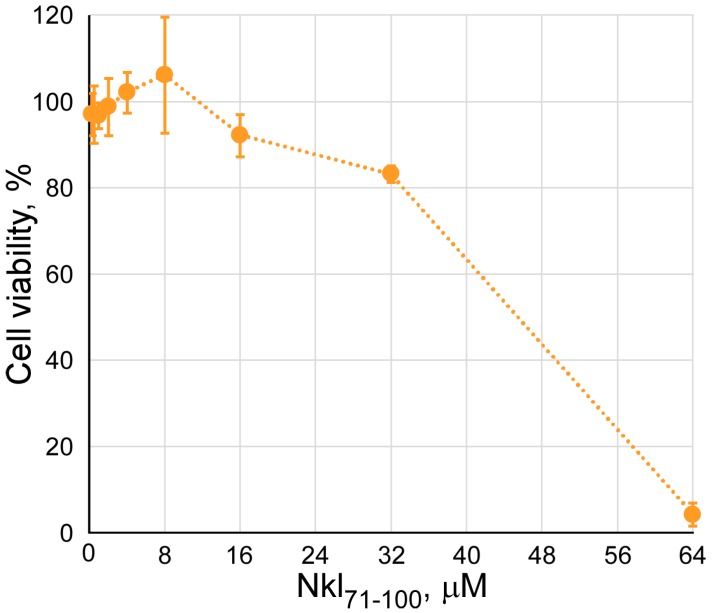
Viability of EPC cells after treatment with Nkl_71–100_. EPC monolayers were treated with increasing concentrations of Nkl_71–100_ for 24 h at 22 °C before performing the MTT assay. Cell viability is shown as the percentage (mean ± s.d.) relative to untreated cells from three independent experiments performed in tetraplicate.

**Figure 5 marinedrugs-17-00087-f005:**
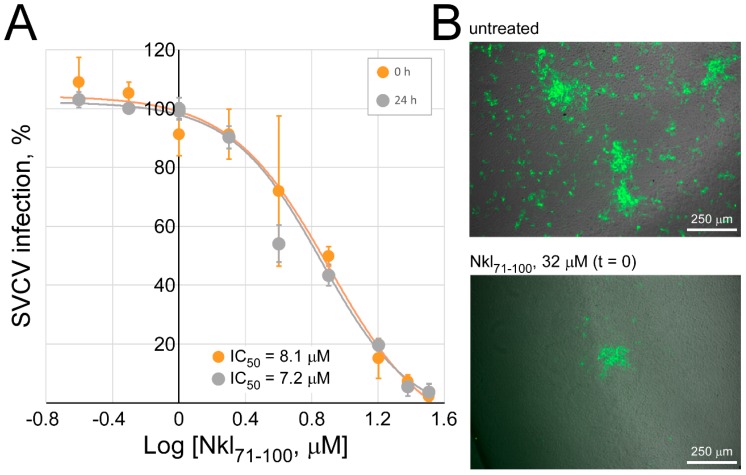
Infectivity of SVCV after preincubation with Nkl_71–100_. EPC monolayers were infected with SVCV (MOI 10^−3^) co-incubated with increasing concentrations of Nkl_71–100_ (0–32 μM) for 24 h. Same mixes performed right before the inoculation to cell monolayers were also included (t = 0). After 20 h of infection, cells were fixed, SVCV-infected foci detected by fluorescent immune-labelling and counted. (**A**) Results are presented as percentage of foci in comparison to those found in corresponding untreated SVCV-infected monolayers, shown as the mean (±s.d.) from two independent experiments performed in triplicate. Solid lines correspond to the best fit to a dose-response curve, from which the IC_50_ values that are included in the graph were calculated. Multiple t test analysis did not give any significant difference between the two datasets. (**B**) Representative images (merging both bright and fluorescent fields) of SVCV-infected cell monolayers with different focus occurrence patterns are shown.

**Figure 6 marinedrugs-17-00087-f006:**
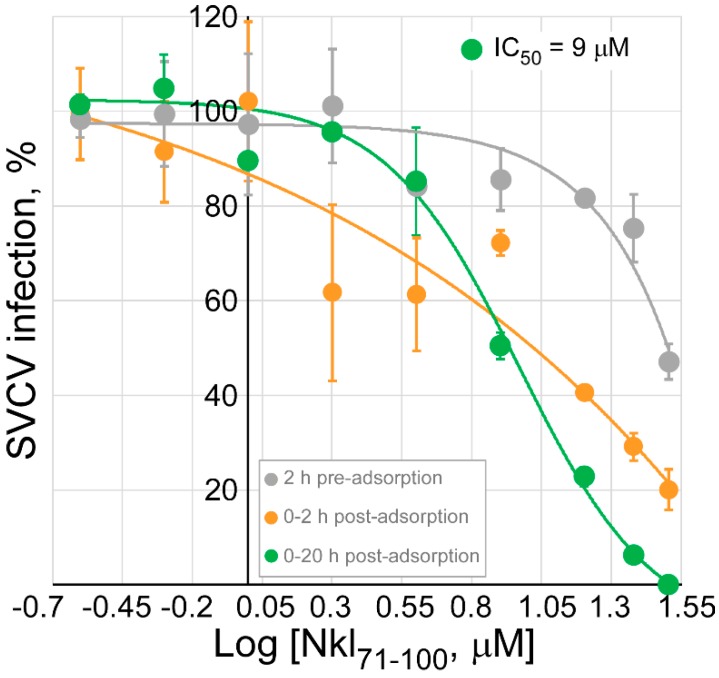
Effect of the timing of the Nkl_71–100_ treatment on the infectivity of SVCV. EPC monolayers were treated 2 h before the inoculation of SVCV (pre-adsorption) and just after the adsorption period for either 2 or 20 h (post-adsorption) with increasing concentrations of Nkl_71–100_ (0–32 μM). Cells were infected with SVCV at MOI 10^−3^. After 20 h of infection, cells were fixed, SVCV-infected foci detected by fluorescent immune-labelling and counted. Results are presented as percentage of foci relative to those found in corresponding untreated SVCV-infected monolayers, shown as the mean (±s.d.) from three different experiments performed in triplicate. Solid lines correspond to the best fit to a dose-response curve, from which the IC_50_ values were calculated, if possible, and included in the graph.

**Figure 7 marinedrugs-17-00087-f007:**
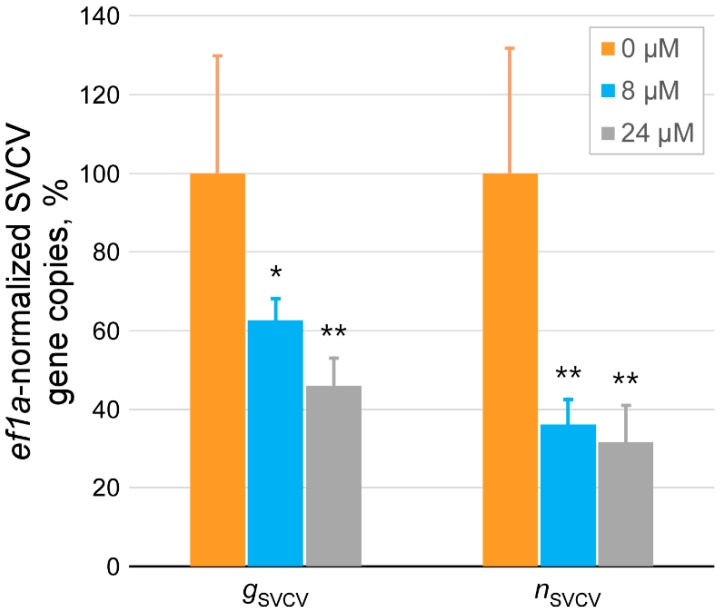
Effect of Nkl_71–100_ on the SVCV attachment to host cell surface. The amount of SVCV particles just adsorbed to host cell in the presence (8 and 24 μM) and absence of Nkl_71–100_ was determined by quantifying their gene copies by RT-qPCR. SVCV was used at MOI 1. Results are shown as percentage of amplified copies (normalized to cellular *ef1a* ones) relative to the value corresponding to untreated monolayers. Each mean (±s.d.) is calculated from three different experiments performed in duplicate. The significance of the changes between treated and untreated groups were indicated as: * *p* < 0.05 and ** *p* < 0.01.

**Figure 8 marinedrugs-17-00087-f008:**
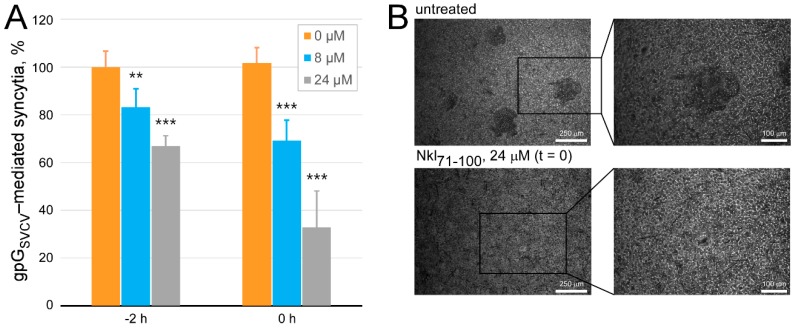
Effect of Nkl_71–100_ on the virus-cell interaction through the SVCV-gpG-mediated fusion activity. The evaluation of the ability of Nkl_71–100_ to inhibit the membrane fusogenic activity of SVCV gpG was determined as previously described by [56]. Treatments included Nkl_71–100_ at 8 and 24 μM which were either incubated with cells for 2 h and then removed prior to the fusion triggering or just added in that moment. **A.** Results are shown as percentage of syncytia relative to that present in corresponding untreated monolayers. Each mean (±s.d.) is calculated from two different experiments performed in triplicate. The significance of the changes between treated and untreated groups were indicated as: ** *p* < 0.01 and *** *p* < 0.001. **B.** Representative bright field images of SVCV-gpG-mediated syncytia taken at two different magnifications.

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
