# Peer review of "Antiviral Activity of a Turbot (Scophthalmus maximus) NK-Lysin Peptide by Inhibition of Low-pH Virus-Induced Membrane Fusion"

_marinedrugs, 2019, doi:10.3390/md17020087_

Reviewer 1 Report

In the manuscript “Antiviral activity of a turbot (Scophthalmus maximus) Nk-lysin peptide by inhibition of low-pH virus-induced membrane fusion” the authors demonstrate that the NK-lysin peptide is able to inhibit SVCV replication by interfering with early steps of replication cycle. The antiviral activity of this short peptide seems related to its ability to inhibit gpG-mediated fusion formation.

The topic of the manuscript is very interesting and deals with the question of drug-resistance and the necessity to discover new broad antiviral drugs. Nevertheless to better clarify the mechanism of inhibition some data should be added.

In Fig. 6 the percentage of SVCV infection shown seems drastically reduced (about 100%) by maintaining the peptide for 20 h after infection. On the contrary, the 2 h treatment is less efficient. Why did the authors exclude the possibility that the peptide may inhibit late steps of infection?

To ascertain this concern, the peptide should be added after 2h from infection (2-20) or late (4-20 and 8-20).

Page 11 lane 376: there is probably a mistake in the text the percentage of infection 20.1 should be referred to 0-2h post adsorption and not pre-adsorption, that should be 47.1 as reported on the graph, Fig 6.

The cell line used is EPC. Since the authors suggest the Nk-lysin peptide as potential antiviral against different human viruses, significance would be higher if human permissive cells were looked at. For example, for rhabdoviruses and herpesviruses (class III): HEK-293 and HeLaS3 cell lines were used (Takacs AM, J Gen Virol 1997; Madhusudana SN, Int J Infect Dis 2010; Marcocci ME, Antimicrob Agents Chemother 2018).

Is the peptide toxic on these cells? Is the mechanism of inhibition the same?

Author Response

Reviewer 1

Comments and Suggestions for Authors

In the manuscript “Antiviral activity of a turbot (Scophthalmus maximus) Nk-lysin peptide by inhibition of low-pH virus-induced membrane fusion” the authors demonstrate that the NK-lysin peptide is able to inhibit SVCV replication by interfering with early steps of replication cycle. The antiviral activity of this short peptide seems related to its ability to inhibit gpG-mediated fusion formation.

The topic of the manuscript is very interesting and deals with the question of drug-resistance and the necessity to discover new broad antiviral drugs. Nevertheless to better clarify the mechanism of inhibition some data should be added. 

In Fig. 6 the percentage of SVCV infection shown seems drastically reduced (about 100%) by maintaining the peptide for 20 h after infection. On the contrary, the 2 h treatment is less efficient. Why did the authors exclude the possibility that the peptide may inhibit late steps of infection?

To ascertain this concern, the peptide should be added after 2h from infection (2-20) or late (4-20 and 8-20).

Response: The reviewer is right and such possibility has not been properly addressed in the text. The following sentence has been added in line 378 in order to clarify this point: “…that Nkl71-100 is at least (since potential inhibitory effects at late steps of infection cannot be discarded yet) affecting viral replication at an early stage in which virus-cell interactions are implicated…”.

We want to thank the reviewer for the experimental suggestion, which will be definitely considered for further studies (in line with those ones raised in the last comment of this review). In this work, however, we wanted to delve into the events happening during the early stages of the infection and their correlation with the results obtained from the biophysical trials.

Page 11 lane 376: there is probably a mistake in the text the percentage of infection 20.1 should be referred to 0-2h post adsorption and not pre-adsorption, that should be 47.1 as reported on the graph, Fig 6.

Response: The reviewer is right. Corresponding values have been changed in the text accordingly.

The cell line used is EPC. Since the authors suggest the Nk-lysin peptide as potential antiviral against different human viruses, significance would be higher if human permissive cells were looked at. For example, for rhabdoviruses and herpesviruses (class III): HEK-293 and HeLaS3 cell lines were used (Takacs AM, J Gen Virol 1997; Madhusudana SN, Int J Infect Dis 2010; Marcocci ME, Antimicrob Agents Chemother 2018).

Is the peptide toxic on these cells? Is the mechanism of inhibition the same?

Response: The reviewer raises an interesting concern. In fact, this same issue is discussed at the end of the manuscript as a hypothesis for future works, in which there will be definitely included the experimental models suggested by the reviewer.

Reviewer 2 Report

The manuscript entitled: “Antiviral activity of a turbot (Scophthalmus maxmus) Nk-lysin peptide by inhibition of low-pH virus-induced membrane fusion”, by Falco et al., describes the antiviral activity of NKl71-100 against SVCV virus. The manuscript is well written, and the experiments are technically adequately done, however few explanations should be included:

1.     The full name of SVCV should be included both in the abstract as well as in the Introduction section.

2.     The EPC cells are grown in 28°C so why the antiviral activity experiments were done in 22°C?

3.     All the antiviral activity experiments were done after treatment with the compound for 24 h, it would be interesting to see the results after 48 h of treatment to see if the antiviral effect is still present.

4.     The authors claim that the compound shows significant antiviral activity however the calculated S.I. (defined as CC50/IC50) is only around 4,9.

Author Response

Reviewer 2

Comments and Suggestions for Authors

The manuscript entitled: “Antiviral activity of a turbot (Scophthalmus maxmus) Nk-lysin peptide by inhibition of low-pH virus-induced membrane fusion”, by Falco et al., describes the antiviral activity of NKl71-100 against SVCV virus. The manuscript is well written, and the experiments are technically adequately done, however few explanations should be included:

1.     The full name of SVCV should be included both in the abstract as well as in the Introduction section.

Response: We agree with the reviewer's comment and the full name of SVCV has been included as requested.

2.     The EPC cells are grown in 28°C so why the antiviral activity experiments were done in 22°C?

Response: The reviewer is right. The optimal temperature for the growth of this cell line is 25-28°C, but they perform well in the range of 20-30°C as stated by the provider (ATCC), and thus the optimal temperature for the replication of SVCV (which is more restricted) was chosen. Actually, EPC cells are commonly used for the replication of other fish rhabdoviruses such as VHSV at even 14°C.

3.     All the antiviral activity experiments were done after treatment with the compound for 24 h, it would be interesting to see the results after 48 h of treatment to see if the antiviral effect is still present.

Response: The reviewer’s suggestion is very interesting and will be definitely considered for follow-up studies aimed at determining the range of such activity. In this work, however, we focused on studying the inhibitory mechanisms occurring at early stages of infection.

4.     The authors claim that the compound shows significant antiviral activity however the calculated S.I. (defined as CC50/IC50) is only around 4,9.

Response: In this case we use the term "significant" in its meaning of "noteworthy" given the experimental design used. In line with the concern previously raised by the reviewer, further studies will be focused on describing the reach of such activity.

Reviewer 3 Report

Submitted paper reported the antiviral activity of a turbot Nk-lysin peptide by inhibition of low-pH virus-induced membrane fusion.

The manuscript was clear and the English provided was professional. Details about experimental section were correctly given. In my view, the manuscript should be accepted as it is.

Author Response

Reviewer 3

Comments and Suggestions for Authors

Submitted paper reported the antiviral activity of a turbot Nk-lysin peptide by inhibition of low-pH virus-induced membrane fusion.

The manuscript was clear and the English provided was professional. Details about experimental section were correctly given. In my view, the manuscript should be accepted as it is.

Response: We are grateful for the comment.

Round  2

Reviewer 1 Report

 The authors have answered to the comments without performing additional experiments. They decided to consider experimental suggestion for another manuscript but in my personal opinion this kind of results would have improved their paper.

Minor revisions have been corrected.